# Global-Scale Assessment of Economic Losses Caused by Flood-Related Business Interruption

Ryo Taguchi [1], Masahiro Tanoue [2], Dai Yamazaki [3,*] and Yukiko Hirabayashi [4]

1   Graduate School of Engineering, The University of Tokyo, Tokyo 113-8656, Japan;
    taguchi@rainbow.iis.u-tokyo.ac.jp
2   Earth System Division, National Institute for Environmental Studies, Tsukuba 300-4352, Japan;
    tanoue.masahiro@nies.go.jp
3   Institute of Industrial Science, The University of Tokyo, Tokyo 153-8505, Japan
4   Graduate School of Engineering and Science, Shibaura Institute of Technology, Tokyo 135-8548, Japan;
    hyukiko@shibaura-it.ac.jp
*   Correspondence: yamadai@rainbow.iis.u-tokyo.ac.jp

**Abstract:** Estimating river flood risk helps us to develop strategies for reducing the economic losses and making a resilient society. Flood-related economic losses can be categorized as direct asset damage, opportunity losses because of business interruption (BI loss), and high-order propagation effects on global trade networks. Biases in meteorological data obtained from climate models hinder the estimation of BI loss because of inaccurate input data including inundation extent and period. In this study, we estimated BI loss and asset damage using a global river and inundation model driven by a recently developed bias-corrected meteorological forcing scheme. The results from the bias-corrected forcing scheme showed an estimated global BI loss and asset damage of USD 26.9 and 130.9 billion (2005 purchase power party, PPP) (1960–2013 average), respectively. Although some regional differences were detected, the estimated BI loss was similar in magnitude to reported historical flood losses. BI loss tended to be greater in river basins with mild slopes such as the Amazon, which has a long inundation period. Future flood risk projection using the same framework under Representative Concentration Pathway 8.5 (RCP8.5) and Shared Socioeconomic Pathway 3 (SSP3) scenarios showed increases in BI loss and asset damage per GDP by 0.32% and 1.78% (2061–2090 average) compared with a past period (1971–2000 average), respectively.

**Keywords:** business interruption loss; flood protection; global scale estimation; future projection

## 1. Introduction

Floods represent a major natural disaster, and flood risk is expected to increase in many regions because of global warming and socioeconomic changes [1–5]. The United Nations Office for Disaster Risk Reduction reported that >2 billion people were affected by floods between 1998 and 2017, constituting 45% of people affected by meteorological disasters such as drought and typhoons [6]. They also reported that total flood-related economic impact during that period was US Dollar (USD) 656 billion. Munich-Re, a reinsurance company in Germany, reported that economic losses from floods in Europe in 2021 reached 54 billion [7].

Flood risk is defined as the expected losses due to a particular flood disaster for a given area and for a certain period [3]. Flood risk comprises hazard (magnitude and/or frequency of flood), exposure affected by flooding, and the system's vulnerability [8], which must be estimated accurately to establish reasonable countermeasures to reduce risks, as it is seen that flood control is a major component of adaptation needs [9]. Because flood disasters occur worldwide and their impact can propagate to the global economic system through supply chain effects, global-scale flood risk assessment has been a focus of recent research. Swiss-Re, a reinsurance company in Switzerland, identified countries that may

experience large economic impacts from future flood events using gross domestic product (GDP) growth and flood risk indices [10]. Mapping flood risk can support decision making in land use planning and flood area management [11]. At the corporate level, such maps can be applied to create business continuity plans that reduce flood risk or minimize flood disaster recovery times [12].

Quantification of disaster risks in previous studies can be classified into damage to infrastructure or buildings (hereafter "asset damage") and opportunity loss caused by business interruption (hereafter "BI loss") [13]. Reports of historical floods published by the Global Facility for Disaster Risk Reduction(GFDRR) have shown that asset damage constitutes a large proportion of total economic losses after natural disasters [14]; moreover, BI loss can be as extensive as asset damage in some instances, such as the 2011 Thailand floods, which caused USD 12 and 13.3 billion (2005 purchase power parity, PPP) in asset and BI loss, respectively. Therefore, it is essential to estimate both asset damage and BI loss because of flood hazard to improve global-scale flood risk assessments.

Previous global flood risk assessments have calculated population and GDP exposure estimates based on flood return periods. Hirabayashi et al. [1,2] projected that greater numbers of people will be affected by floods with return times of >100 years under a warmer climate. Ward et al. [15] estimated an annual GDP risk of USD 1383 billion (2005 PPP) by calculating flood hazard, exposure, and vulnerability on a global scale [16]; they overlaid GDP exposure and inundation depth hazard, then calculated the proportion of damage using a depth–damage function to represent vulnerability. A more realistic estimation of global flood risk was recently proposed, considering the level of flood protection required for each region [17]. However, these previous studies solely focused on asset damage [15,16,18], and model-based estimation of BI loss due to flood was limited to limited regions [19].

Previous studies have estimated regional-scale BI loss based mainly on local surveys or questionnaires regarding flood damage. For example, BI loss was estimated by Thieken et al. [20] for a 2013 flood in Germany and by Yang et al. [21] for the September 2000 Tokai heavy rain in Japan. Business interruption data are typically collected through surveys that involve companies directly affected by floods; thus, it is difficult to expand the same methodology to a global scale. It is also difficult to estimate BI loss using the methods typically applied to estimate global asset damage, which focus on the maximum flood stage without considering flood duration. To estimate BI loss, a business stagnation period should be calculated based on simulated flood duration. Flood risk models could potentially simulate reasonable daily inundation depth; however, this process is complicated by bias in the meteorological forcing data. Therefore, previous studies have estimated global asset damage by directly applying bias correction to the annual maximum inundation depth [16].

The recent release of the bias-corrected meteorological forcing data product S14FD [22] has enabled direct simulation of daily inundation depth. Using these data, Tanoue et al. [19] showed that modeled inundation depth and inundation period were reasonably correlated with observations for the 2011 Thailand flood; these findings provided estimates of asset damage-related direct economic loss that were similar to estimates from the World Bank. Tanoue et al. [19] also demonstrated that BI loss for the 2011 Thailand flood was greater than direct economic damage caused by the flood, because of the shallow slope of the Chao Phraya River. In the present study, we used the approach established by Tanoue et al. [19] in terms of the same modeling framework; we expanded the approach to include global river basins, thereby estimating flood-related global asset damage and BI loss for present and future scenarios to explore potential changes in flood risk impact. Please note that in this study we mainly focused on BI loss and compared its importance relative to asset damage which have been well studied in previous studies [15,16]. We did not estimate high-order propagation effects (Shughrue et al. [23]), since estimating this effect requires us to use Computable General Equilibrium (CGE) and the data used for CGE are limited.

## 2. Materials and Methods

### 2.1. Overview

In this study, we calculated BI loss and asset damage based on inundation period and inundation depth data derived from flood simulations and a gridded GDP dataset. We defined BI loss as the loss of opportunity because of interrupted business activity; we defined asset damage as direct economic damage, such as the destruction of physical assets. A summary of the BI loss and asset damage estimation process is presented in Figure 1. We determined the occurrence of inundation when the magnitude of modeled daily total storage exceeded the local flood protection standard for each grid. BI loss and asset damage were estimated at a scale of $30'' \times 30''$ (ca. 1 km × 1 km at the equator) and aggregated to geographical units such as basins, countries, and continents.

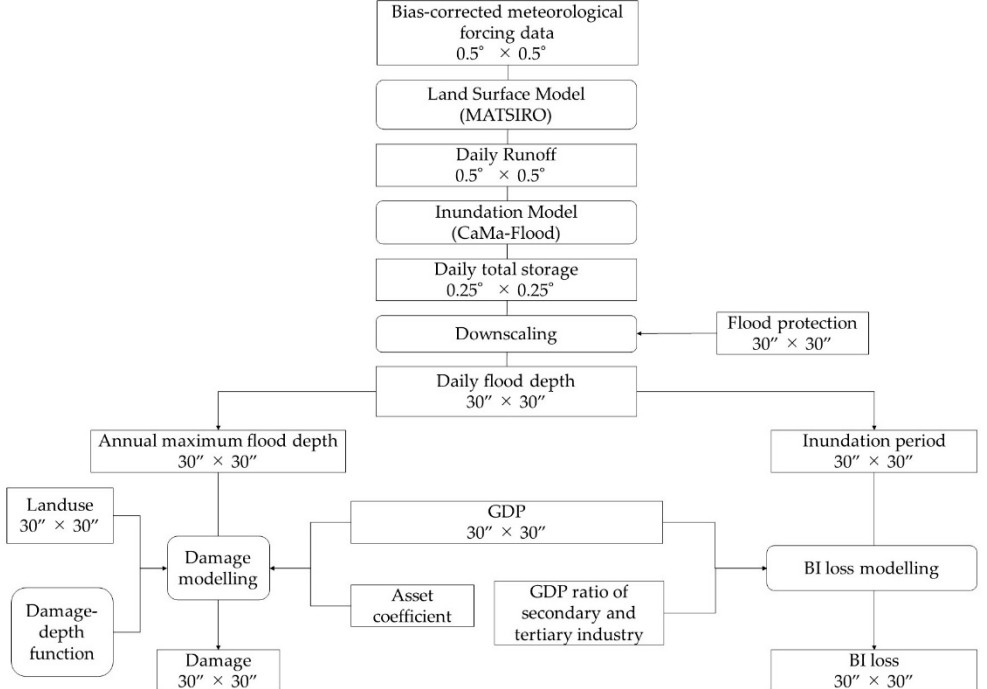

**Figure 1.** Flowchart of the estimation method of BI loss and asset damage.

In this study, we used bias-corrected meteorological forcing data. The calculation of inundation period and BI loss requires daily inundation extent and depth data; these variables were not examined in previous global flood risk studies [1,3,15], which focused on extreme values (e.g., values for 100-year floods, as defined by annual maximum discharge or flood water volume). In such studies, biases in discharge or flood water volume caused by meteorological forcing errors were corrected by applying the rate of change in discharge or flood water volume from reanalysis products. Therefore, we anticipate that the application of bias-corrected atmospheric forcing in this study will improve the representation of river hydrodynamics simulations, including daily inundation depth and extent. Tanoue et al. [19] implemented the Catchment-based Macro-scale Floodplain (CaMa-Flood) [24] model using this bias-corrected weather scheme; they detected similar interannual and annual variations, as well as similar inundation extent and duration, between modeled discharge and in situ observations.

### 2.2. Model

The methods and models used for flood simulation in this study were described in detail by Tanoue et al. [19]. Briefly, we used CaMa-Flood [24] to calculate daily total water storage and inundation depth. CaMa-Flood explicitly calculates river discharge, water depth, and flood extent at each model time step on a global scale by assuming complex

floodplain inundation dynamics as the sub-grid physics. CaMa-Flood is implemented by external runoff forcing (typically calculated by rainfall–runoff or land surface models). The simulation is conducted at a coarse resolution (in this study, 0.25°, ca. 25 km at the equator) to achieve efficient global-scale analysis; however, the simulated flood depth can be diagnostically downscaled using high-resolution topography data in post-processing (in this study, 30″, ca. 1 km).

Runoff input for CaMa-Flood was calculated using Minimal Advanced Treatments of Surface Interaction and RunOff (MATSIRO) [25], which is a land surface component of the Japanese climate model MIROC; it calculates water and energy balance on the land surface (including runoff from land to river, which is used as input for CaMa-Flood), using weather forcing as input. Previous studies confirmed that the combination of CaMa-Flood and MATSIRO runoff data reasonably simulates river discharge and water level along major rivers [24]. Results from CaMa-Flood and MATSIRO have been used for global flood risk analysis in many studies [1,2,4,18,25–31].

### 2.3. Data

#### 2.3.1. Atmospheric Data

In this study, we used a reduced-bias meteorological forcing dataset obtained from the S14FD reanalysis dataset (S14FD-Reanalysis) and bias-corrected global climate model (GCM) output forcing dataset (S14FD-GCM) [22], which allowed the use of daily model output values without bias correction. S14FD-Reanalysis is a forcing dataset that contains global data for 1958–2013 with a grid size of 0.5° and three time resolutions (3-hourly, daily, and monthly). S14FD-Reanalysis is based on the Japanese 55-year Reanalysis (JRA-55) product [32], with each variable corrected using observation datasets such as CRU-TS3.22 [33] and GPCCv7 [34]. For example, precipitation is corrected for monthly variation using GPCCv7 data and CRU-TS3.22 data; it is re-calculated using corrected variables such as temperature and wind speed.

The S14FD-GCM dataset includes 11 forcing variables from eight GCMs: GFDL-ESM2M, IPSL-CM5A-LR, MIROC-ESM-CHEM, HadGEM2-ES, NorESM1-M, MIROC5, MIROC-ESM, and MRI-CGCM3 [22]. The GCM dataset was constructed from five GCMs: GFDL-ESM2M, HadGEM-ES2, IPSL-CM5A-LR, MIROC-ESM-CHEM, and NorESM1-M. The S14FD-GCM dataset contains historical (1961–2005) and future (2006–2100) data.

These datasets represent observed extreme temperature and precipitation data more accurately than do other GCM forcing data. According to Tanoue et al. [19], the absolute values of annual maximum inundation in these datasets correspond to historical flood data. Using the reduced-bias meteorological forcing dataset, we obtained daily water storage data to calculate inundation periods and BI losses.

#### 2.3.2. Geographical Data

To determine asset damage in urban areas, we used current land use data from the Global Land Cover by National Mapping Organizations (GLCNMO) v2 product [35] to calculate the percentage of urban area of each grid. GLCNMO recognizes 20 land use categories and has a horizontal resolution of 15″ × 15″. We calculated the percentage of urban area at 30″ × 30″ horizontal resolution from GLCNMO.

We used national mask data derived from the Global Rural-Urban Mapping Project v1 (GRUMPv1) [36] to aggregate asset damage or BI loss at the country scale. We applied sub-country administrative units used in the flood protection standard dataset FLOPROS [17] as a subnational mask. The subnational mask was used to aggregate subnational building heights and GDPs as described below. The simulation results were also analyzed on the sub-basin scale; therefore, we created a sub-basin mask by dividing tributaries with drainage area > 10,000 km$^2$.

### 2.3.3. Socioeconomic Data

National GDP data were used to estimate gridded GDP values of secondary and tertiary industries and their assets. Although the National Institute for Environmental Studies publishes global GDP data [37], their horizontal resolution is $0.5° \times 0.5°$, which is insufficient for flood damage estimation. Previous studies involving flood damage estimation have calculated gridded GDP via multiplication of the GDP per capita of each country by the gridded population data [13,34]. However, this approach assumes that GDP per capita is homogenous at the country scale; it does not reflect differences between urban and rural regions [38]. Therefore, in this study, we assumed that building height within each grid was proportional to GDP per capita [39,40]; we derived gridded GDP according to the weighted distribution of country GDP with respect to building height data at finer resolution, as previous studies showed that building height is proportional to GDP per capita in European cities [33,34].

Country GDP data were derived from GDP per capita (USD, 2005 PPP) and the total population of each country. These datasets were obtained from World Bank Open Data (available online: https://data.worldbank.org/ (accessed on 17 January 2022)) for 1960–2013 (past simulation) and 2014–2100 (future simulation for SSP3). Building height data [41] were prepared at $30'' \times 30''$ resolution using the Advanced Spaceborne Thermal Emission and Reflection Radiometer global digital elevation model.

We used data from 11 countries, constituting 319 subnational data in total (Table 1), to obtain the relationship between building height and gridded GDP, which we then applied globally. We searched GDP data including subnational GDP from the government or other public institutions of each country and only these countries are available. The equation for the relationship between building height and gridded GDP was developed as follows. First, we estimated the relationship between the ratio of subnational GDP to country GDP (the subnational GDP ratio, $s\_GDP_{ratio}$) and the total building height within each subnational region to the total building height within the country (subnational building ratio, $s\_bld_{ratio}$) under the assumption that this relationship was linear:

$$s\_GDP_{ratio} = 0.9634 \times s\_bld_{ratio} + 0.0011 \tag{1}$$

**Table 1.** Sources of subnational GDP for each country.

| Country | Year | Data Source | Reference |
|---|---|---|---|
| United States | 2017 | Bureau of Economic Analysis | Available online: https://apps.bea.gov/itable/iTable.cfm?ReqID=70&step=1#reqid=70&step=1&isuri=1 (accessed on 17 January 2022) |
| India | 2013 | Ministry of Statistics and Programme Implementation | Available online: https://www.mospi.gov.in/download-tables-data (accessed on 17 January 2022) |
| Australia | 2016 | Australian Bureau of Statistics | Available online: https://ipfs.io/ipfs/QmXoypizjW3WknFiJnKLwHCnL72vedxjQkDDP1mXWo6uco/wiki/List_of_Australian_states_and_territories_by_gross_state_product.html (accessed on 17 January 2022) |
| Canada | 2015 | Statistics Canada | Available online: https://www150.statcan.gc.ca/t1/tbl1/en/tv.action?pid=3610048701 (accessed on 17 January 2022) |

**Table 1.** *Cont.*

| Country | Year | Data Source | Reference |
|---|---|---|---|
| Thailand | 2010 | National Economics and Social Development Board | Available online: https://www.nesdc.go.th/nesdb_en/main.php?filename=index (accessed on 17 January 2022) |
| Germany | 2015 | Federal Statistical Office | Available online: http://www.de-info.net/kiso/laenderbip.html (accessed on 17 January 2022) |
| Brazil | 2015 | Brazilian Institute of Geography and Statistics | Available online: https://agenciadenoticias.ibge.gov.br/agencia-sala-de-imprensa/2013-agencia-de-noticias/releases/17999-contas-regionais-2015-queda-no-pib-atinge-todas-as-unidades-da-federacao-pela-primeira-vez-na-serie (accessed on 17 January 2022) |
| China | 2009 | Chinese Statistical yearbook | Available online: http://www.spc.jst.go.jp/statistics/stats2010/ (accessed on 17 January 2022) |
| Japan | 2014 | Cabinet Office, Government of Japan | Available online: http://www.esri.cao.go.jp/jp/sna/data/data_list/kenmin/files/contents/main_h26.html (accessed on 17 January 2022) |
| South Africa | 2010 | Statistics South Africa | Available online: https://ipfs.io/ipfs/QmXoypizjW3WknFiJnKLwHCnL72vedxjQkDDP1mXWo6uco/wiki/List_of_South_African_provinces_by_gross_domestic_product.html (accessed on 17 January 2022) |
| Chile | 2017 | Central Bank of Chile | Available online: https://si3.bcentral.cl/siete/EN/Siete/Cuadro/CAP_CCNN/MN_CCNN76/CCNN2013_P0_V2 (accsessed on 17 January 2022) |

Using this relationship, we obtained a maximum correlation coefficient of 0.91 for a lower building height limit of 6.0 m. Next, we derived the ratio of gridded GDP to country GDP (gridded GDP ratio, $g\_GDP_{ratio}$) from the ratio of gridded building height to total building height within the country (gridded building height ratio, $g\_bld_{ratio}$) and the relationship between $s\_GDP_{ratio}$ and $s\_bld_{ratio}$. For a building height of 0, we converted $g\_GDP$ to 0. Finally, we converted $g\_GDP_{ratio}$ such that the total GDP ratio per country was equal to 1.

$$g\_GDP = g\_GDP_{ratio} \times \text{country GDP}$$
$$= (0.9634 \times g\_bld_{ratio} + 0.0011) \times \text{country GDP} \qquad (2)$$

In the absence of a relevant dataset for secondary and tertiary industries, their GDP ratios were derived by subtracting the percentage of primary industry GDP, derived from 2010 World Bank Open Data values, from 1. For countries with missing data, we used the global average.

We assumed that assets would be 2.8 × GDP in urban areas [42] and equal to GDP in other areas [16]. The factor 2.8 was derived from the relationship between GDP per

capita and capital produced per capita in urban areas. The urban area was calculated using GLCNMO, as described in Section 2.3.2.

### 2.4. Calculation of Inundation Period and Annual Maximum Inundation Depth

We calculated inundation period and annual maximum inundation depth at $30'' \times 30''$ resolution (ca. 1 km × 1 km at the equator) from calculated daily water storage at $0.25° \times 0.25°$ (ca. 25 km × 25 km at the equator) using the FLOPROS dataset [17] and a high-resolution digital elevation model. The method was described in detail by Tanoue et al. [19]. Briefly, to reflect the effects of current flood protection standards, we calculated the overflow flood water depth at $0.25° \times 0.25°$ resolution from calculated daily total storage and total storage data corresponding to current flood protection standards obtained from FLOPROS. Inundation was calculated only when the calculated daily total storage exceeded the total storage, which is equivalent to the current flood protection standard. Total storage corresponding to the current flood protection standard was calculated using the Gumbel distribution with L-moment methods from the annual maximum total storage for 1961–2005. The overflow flood water depth was downscaled to a horizontal resolution of $30'' \times 30''$ using a high-resolution digital elevation model. The flooded area fraction was calculated at the same resolution.

### 2.5. Estimation of BI Loss and Asset Damage
#### 2.5.1. BI Loss Estimation

BI loss was categorized into an indirect and consequential effect [43] and defined as loss caused by the inability of people to perform work due to workplace destruction or inaccessibility [44]. In this study, BI loss was calculated from the inundation period, gridded GDP, and the GDP ratio of secondary and tertiary industries.

We assumed that no daily production occurred during inundation periods; we defined the BI period as a period of linear recovery after inundation. Thus, BI loss was calculated as the difference in secondary and tertiary industry production between non-flood and flood periods, as follows:

$$\text{BI loss} = \left( \text{IP} + \frac{\text{BIP}}{2} \right) \times \frac{\text{AP}}{\text{Nd}} \tag{3}$$

$$\text{BIP} = \text{IP} \times a \tag{4}$$

where IP is the inundation period, BIP is the BI period, AP is the annual production of secondary and tertiary industries, and Nd is the number of days per year. For each country, the daily production of secondary and tertiary industries was obtained by dividing GDP by the number of days in the corresponding year (assuming constant daily production), then multiplying the result by the GDP ratio of secondary and tertiary industries for each country.

The BI period was assumed to be proportional to the inundation period (Equation (4)). The proportional constant $\alpha$ is difficult to obtain on a global scale because it varies regionally [19] and the available examples are limited. There are three available examples which estimated this constant in a survey of past flood (Japan, USA, and Thailand). In Japan, the BI period is twofold longer than the inundation period according to the results of a survey on flood events in Japan [45]. In the USA, a restaurant hit by a typhoon was closed for 84 days because of inundation, then returned to its pre-typhoon state 211 days after its closure, which is 2.5-fold longer than the inundation period. In Thailand 2011 floods, BI period is more than 10 times longer than inundation period which is estimated based on analysis of interview reports [19]. As mentioned before, this constant differs from region to region and the estimation methods. In this study, we determined this constant as two based on the above surveys of past floods. We note that applying the results of these three cases globally is a very simplistic assumption, and this is recommended to be revised when more data becomes available.

2.5.2. Asset Damage Estimation

In this study, asset damage was defined as the damage caused by the most severe flood in each year in order to make a fair comparison to previous studies which typically estimated it by annual maximum flood depth [10,14,17]. Asset damage was calculated through the multiplication of asset value by the fraction of assets with flood-related damage. Assets were calculated through the multiplication of GDP by the asset coefficient, following a common practice in previous studies. In this study, we used the annual maximum inundation depth to calculate asset damage for the most severe flood in each year. The damage fraction was derived using a damage–depth function proposed by Huizinga et al. [46], which incorporates diverse regions and sectors that experience flood damage. These regions are classified into the following continents, based on geographical classification by the World Bank and Dottori et al. [47]: Europe, North America, Central and South America, Asia, Africa, and Oceania. The sectors included residential, commercial, industrial, transport, infrastructure, and agriculture sectors. In this study, we averaged residential, commercial, and industrial damage for each inundation depth and region.

*2.6. Validation Dataset*

We used the Emergency Events Database (EM-DAT, Available online: http://www.emdat.be (accessed on 17 January 2022)) and World Bank data to validate our estimation methods. EM-DAT contains occurrence and effects data for >22,000 disasters worldwide from 1900 to the present day, based on data from UN agencies, non-governmental organizations, insurance companies, research institutes, and press agencies.

The World Bank commissioned the Damage, Loss and Needs Assessment guidance notes to quantify and understand the social, economic, and financial impacts of disasters [48]. The Global Faculty for Disaster Risk Reduction website lists approximately 60 reports; of these, we used reports that described flood events occurring during our calculation period and applied the same definition of BI loss as the present study. These targeted flood events are summarized in Table 2. The targeted industries were secondary and tertiary industries based on the manual for economic evaluation of flood control investments [45].

**Table 2.** Overview of targeted flood events.

| Country | Year | Loss (USD 2005 PPP) | Targeted Industry | Characteristics |
|---------|------|---------------------|-------------------|-----------------|
| Namibia | 2009 | 56 million | Infrastructure, Industry, Commerce, Tourism | Floods were caused by heavy rainfalls and exacerbated by drainage system that were unable to handle the volumes of water |
| Moldova | 2010 | 5.34 million | Energy, Roads, Railways, Water and Sanitation | Heavy rainfalls breached dams. The overall situation improved slowly since repairing dams was delayed and outflow from inundation area back to river was limited. |
| Pakistan | 2010 | 1.21 billion | Transport and Communications, Water Supply and Sanitation, Energy, Private Sector and Industries, Financial Sector | Heavy rainfalls in monsoon season caused landslide and flash flood which broke major embankments and canals. |
| Pakistan | 2011 | 121 million | Transport and Communications, Water Supply and Sanitation. Energy, Private Sector, Industries and Financial Sector | Heavy rainfalls in monsoon season caused flash flood. |

**Table 2.** *Cont.*

| Country | Year | Loss (USD 2005 PPP) | Targeted Industry | Characteristics |
|---------|------|---------------------|-------------------|-----------------|
| Thailand | 2011 | 13.3 billion | Transport, Telecommunication, Electricity, Water Supply and Sanitation, Manufacturing, Tourism, Finance and Banking | Heavy rainfalls overflowed or breached 10 major flood control systems. |
| Malawi | 2012 | 0.60 million | Transport, Water and Sanitation | Heavy rainfall caused flood twice. |
| Nigeria | 2012 | 1.65 billion | Manufacturing, Commerce, Oil, Electricity | Heavy rainfalls caused river flood which breached irrigation reservoirs. |

Losses estimated by EM-DAT or the World Bank are typically expressed in terms of nominal GDP, in local currency. We converted these values to USD (2005 PPP) to match the currency of estimated BI loss, but this conversion was difficult because of missing data. Therefore, we calculated the ratio of estimated loss with respect to the nominal GDP for each country (derived from World Bank Open Data) and multiplied the result by GDP (2005 PPP).

*2.7. Simulation Conditions*

In this study, we validated the estimation method and analyzed past floods by conducting a reanalysis simulation, then compared past and future flood impacts on the economy by conducting a GCM simulation. For the reanalysis simulation, we used the S14FD-Reanalysis forcing dataset and past GDP data for 1960–2013. For the GCM simulation, we used a historical S14FD-GCM (1961–2005) dataset and GDP data to estimate past flood impacts on the economy; we used a future S14FD-GCM dataset for Representative Concentration Pathways 8.5 (RCP8.5) (2006–2098) and GDP data for Shared Socioeconomic Pathway 3 (SSP3) to estimate future flood impacts on the economy.

**3. Results**

*3.1. Reproducibility of Economic Loss Caused by Past Floods*

We compared simulated asset damage and BI loss to reported values, with the aim of confirming that CaMa-Flood was able to reproduce the general trend of global flood risk. Globally, annual total BI loss and asset damage averaged from 1960 to 2013 were estimated to be USD 26.9 and 130.9 billion (2005 PPP), respectively, in the reanalysis simulation. To validate the method for estimating BI loss and asset damage with flood protection, we compared the estimation results with reported values and previous studies, respectively.

We compared our simulated past BI loss to BI loss reported in the World Bank historical flood data described in Section 2.6 (Figure 2). The comparison results include error related to BI loss without considering flood protection in some countries. Estimated BI losses calculated for Namibia, Moldova, Pakistan (2011), Thailand, and Malawi were similar to the World Bank values. In other countries, estimated BI losses including flood protection were smaller than World Bank values.

To validate the methodology of asset damage estimation, we compared calculated global asset damage with values obtained from previous studies [3,15]. Ward et al. [15] estimated annual global asset damage averaged from 1960 to 1999 at USD 94 billion (2005 PPP), whereas the annual global asset damage estimated in this study was USD 96 billion (2005 PPP) for the same period. Alfieri et al. [3] estimated that the annual global asset damage averaged from 1976 to 2005 in targeted river basins with drainage area > 5000 km$^2$ was USD 65 billion (2005 PPP), whereas the annual global asset damage estimated in this study was USD 56.7 billion (2005 PPP) for the same period and conditions. Therefore, we

conclude that our method of calculating asset damage was sufficiently accurate, at least for global long-term averages.

The spatial distribution of annual BI loss and BI loss per GDP averaged from 1960 to 2013 according to river basin drainage area ($\geq$10,000 km$^2$) are shown in Figure 3.

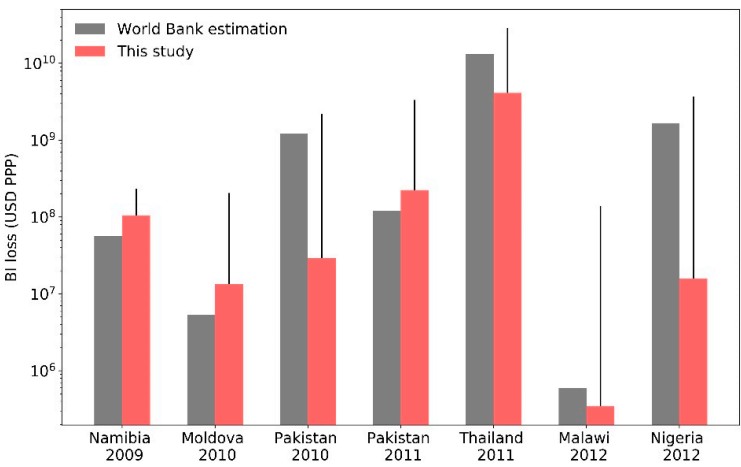

**Figure 2.** Comparison of simulated BI loss (red bars) with World Bank values (gray bars). Data are shown on a logarithmic scale; error bars indicate uncertainties related to the inclusion of BI loss data that do not consider flood protection.

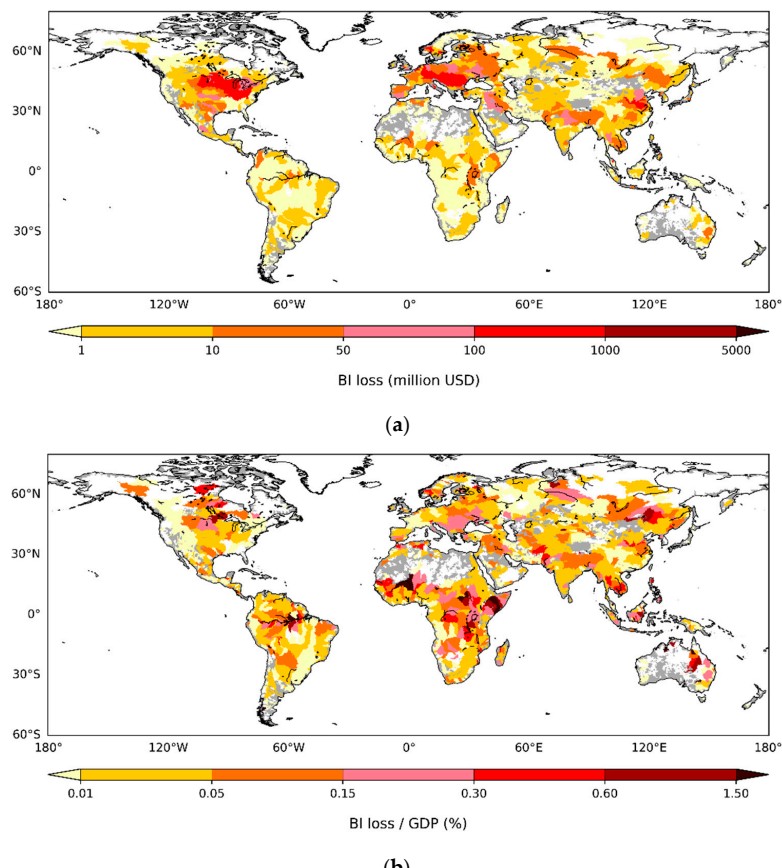

**Figure 3.** (**a**) BI loss. (**b**) Percentage of BI loss per GDP in river basins with drainage area $\geq$ 10,000 km$^2$, averaged from 1960 to 2013. River basins with area < 10,000 km$^2$ are indicated in gray and minimal BI loss are indicated in white.

BI loss exceeded USD 50 million (2005 PPP) in drainage areas of America and Europe, as well as the Tigris, Euphrates, Chao Phraya, and Yellow Rivers, where GDP was high and slopes were mild (Figure 3a). In America and Europe, high GDP contributed to high BI loss. In drainage areas of the Tigris, Euphrates, Chao Phraya, and Yellow Rivers where slopes were mild, inundation periods were prolonged, resulting in high BI loss. BI loss was USD < 1 million (2005 PPP) in most parts of South America and Africa, where GDP was USD < 50 billion (2005 PPP) in most targeted river basins.

To remove the effects of economic scale, we calculated BI loss as a percentage of GDP for each basin (Figure 3b). The Amazon, Congo, Shebelle, Niger, Indus, Diamantina and Chao Phraya River basins had values > 0.3% of GDP (i.e., more than double the global average of 0.12%). These areas had long inundation periods because of their shallow slopes. For example, the Niger inland delta is completely inundated for long periods in the rainy season [49]. In contrast, America and Europe had greater absolute BI loss than other regions (Figure 3a), but they showed a relative BI loss of <0.1% of GDP, which was smaller than the global average. The economic impact of floods in these regions was smaller because of their high GDP; thus, the ratio of BI loss to GDP was smaller than in countries with lower GDP.

### 3.2. Assessment of Climate Change Impact on Flood-Related Economic Loss

We projected future BI loss and asset damage using bias-corrected S14FD-GCM forcing data, S14FD-Reanalysis data, and output from CaMa-Flood driven by S14FD-GCM. We used average values produced by five GCMs to compensate for uncertainties in each GCM.

We compared changes in the percentage of BI loss to GDP over time with changes in asset damage relative to GDP according to S14FD-Reanalysis and S14FD-GCM data (Figure 4).

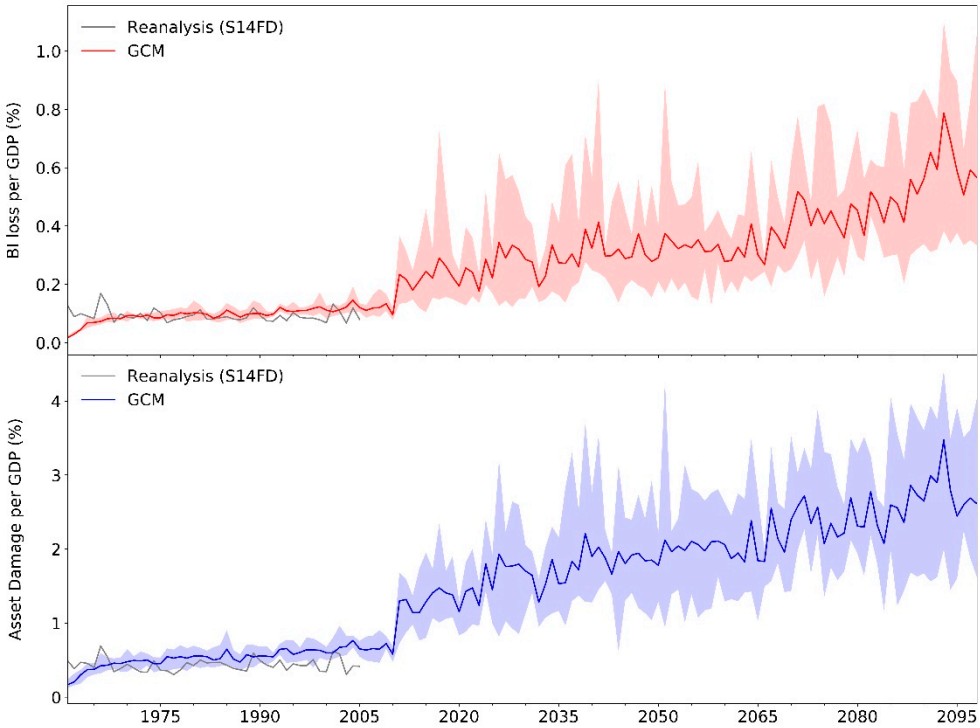

**Figure 4.** Changed in flood-related global annual BI loss (red line) and asset damage (blue line) from 1961 to 2098. Shaded areas indicate the range of maximum and minimum values of S14FD-GCM simulation results. Gray line indicates S14FD-Reanalysis product values.

First, to validate the results from GCM simulation, we compared the calculation result of BI loss and asset damage from historical GCM simulation (available for 1961–2005) against data from the reanalysis simulation. We assumed that the reanalysis simulation was expected to have smaller uncertainty because its forcing data was constrained by past meteorological observations, while GCM simulations were based on model experiments. Global total annual BI loss from reanalysis simulation averaged through 1961 to 2005 was USD 23.1 billion (2005 PPP), whereas average total annual BI loss from the GCM simulation was USD 25.8 billion (2005 PPP). The maximum variation of each GCM from the average value was 33.0%. In contrast, global total annual asset damage and asset damage per GDP from S14FD-Reanalysis was USD 109.7 billion (2005 PPP), whereas the average global total annual asset damage and asset damage per GDP from the GCM simulation was USD 142.5 billion (2005 PPP). The maximum variation of each GCM from the average value was 43.2%. The results from reanalysis simulation and GCM simulation of BI loss and asset damage were similar between simulations. Therefore, we concluded that averaged BI loss and asset damage from the GCM simulation were sufficiently accurate for estimating future BI loss and asset damage.

Next, we estimated future BI loss and asset damage for the future (near-term 2021–2050 and long-term 2061–2090) and compared them with the estimates for the past period (1971–2000). The results are summarized in Table 3. Among all GCMs, the maximum deviations from the average BI loss and asset damage were 137.9% and 101.2%, respectively. Global annual BI loss and asset damage were projected to increase by 34.6- and 34.7-fold, respectively, from 1971–2000 to 2061–2090. Global annual BI loss and asset damage per GDP during 2061–2090 are projected to be 4.2- and 4.2-fold greater, respectively, than those values during 1971–2000, indicating that increases in BI loss and asset damage are anticipated because of climate change.

**Table 3.** GCM simulation results.

| | Annual Average (Billion USD (2005 PPP)) | | | Annual Average per GDP | | |
|---|---|---|---|---|---|---|
| | 1971–2000 | 2021–2050 | 2061–2090 | 1971–2000 | 2021–2050 | 2061–2090 |
| BI loss | 27.2 | 435.5 | 940.4 | 0.10% | 0.29% | 0.42% |
| Asset damage | 150.0 | 2586.1 | 5201.3 | 0.55% | 1.74% | 2.33% |

Comparisons of the spatial distributions of past (1971–2000) and future (2061–2090) BI loss and percentage of BI loss relative to GDP are shown in Figure 5. The target river basins were basins with area $\geq$ 10,000 km$^2$. The average Bl loss and percentage of BI loss relative to GDP in the targeted river basins were USD 951.7 million (2005 PPP) and 0.81%, respectively. Future BI loss was projected to exceed USD 5 billion (2005 PPP) in most parts of China, Thailand, and the Nile River, representing an increase of USD > 1 billion (2005 PPP) relative to past values. In most parts of America and Australia, BI loss was projected to increase by USD <100 million (2005 PPP), despite their economic growth. The percentage of BI loss relative to GDP was projected to exceed 1.5% in the part of Nile and Niger Rivers and some river basins in Sub-Sahara, Indonesia, and Thailand. The percentage of BI loss relative to GDP is expected to increase by >1.0% in drainage areas of the Nile and Niger Rivers; it is expected to decrease in most regions of South America, the Middle East, and Australia.



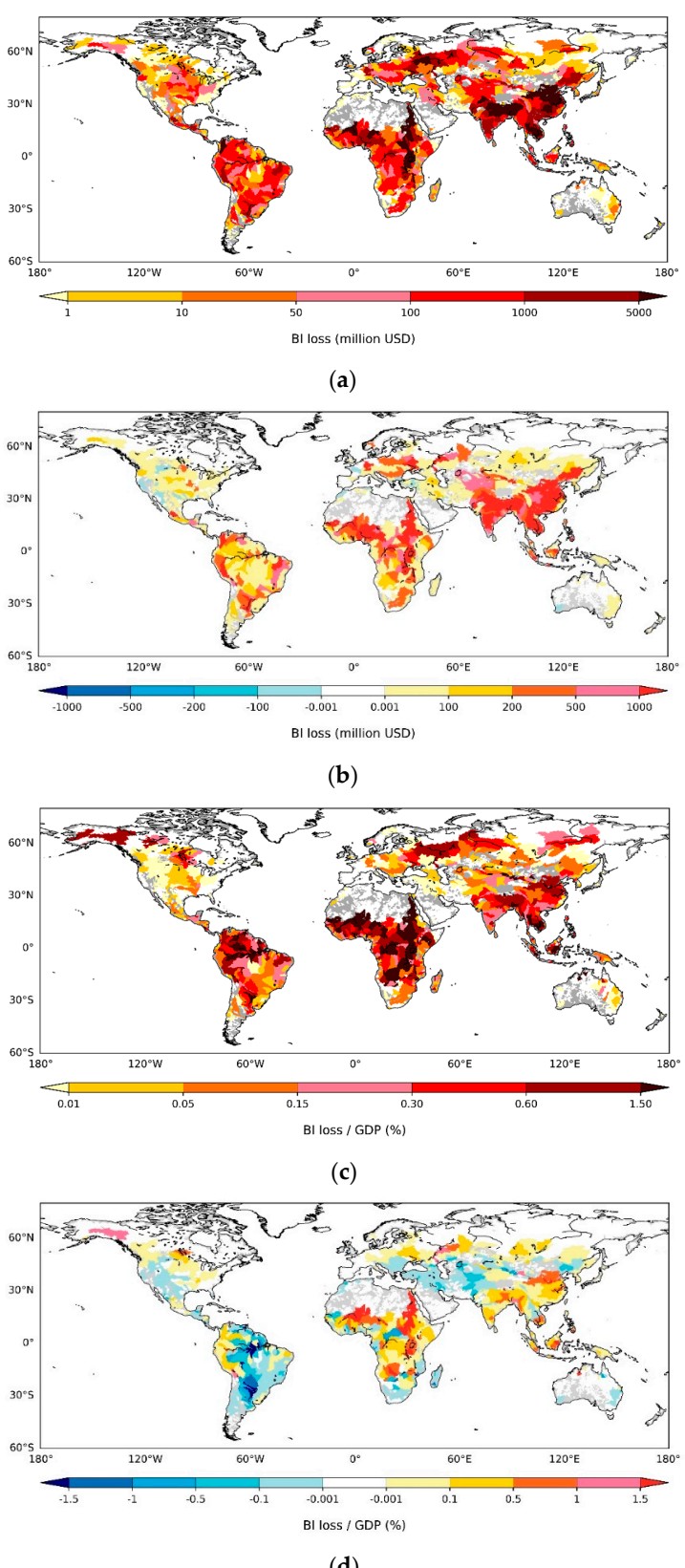

**Figure 5.** (**a**) Future BI loss (2061–2090). (**b**) Difference in BI loss between the future and past (1971–2000) study intervals. (**c**) Future percentage of BI loss relative to GDP. (**d**) Difference in the percentage of BI loss relative to GDP between future and past study intervals. Gray area indicates basins with area < 10,000 km². White area indicates basins with minimal BI loss or percentage of BI loss relative to GDP.

## 4. Discussion

### 4.1. Reanalysis Estimation vs. World Bank Estimation

As shown in Figure 2, BI losses estimated in this study were smaller than BI loss reported by the World Bank for historical floods in various countries. In Pakistan, landslides were triggered by heavy rain and floods in 2010, causing severe damage to infrastructure. In the present study, we defined BI loss as loss related to business interruption caused only by floods, which resulted in underestimation. In 2012, severe floods in Nigeria drastically reduced productivity in commercial and industrial properties located near the Niger River, resulting in BI loss comparable to asset damage. Reduced productivity at these companies influenced productivity in other areas. In particular, the oil sector experienced greater flood impacts because of losses in associated sectors such as infrastructure. We did not include such indirect losses in our BI loss estimates, unlike Carrera et al. [50]; therefore, our BI loss estimates were smaller than the estimates from the World Bank.

### 4.2. Comparsion of Estimated BI Loss against Estimated Asset Damage

Previous asset damage-focused studies underestimated the economic impacts of floods, particularly in regions where the percentage of BI loss relative to asset damage was high. We compared the relative impacts of BI loss and asset damage on regional economies using reanalysis simulation results. Globally, the ratio of BI loss to asset damage averaged from 1960 to 2013 was 20.6%. This value was highest in African regions (22.1%) and Europe (26.9%), while it was lowest in South America (6.1%). The ratio of BI loss to asset damage for river basins with area $\geq$10,000 km$^2$ is shown in Figure 6. The ratio of BI loss to asset damage exceeded 100% in some areas, indicating that its influence on economic damage cannot be ignored in flood risk estimation for these areas.

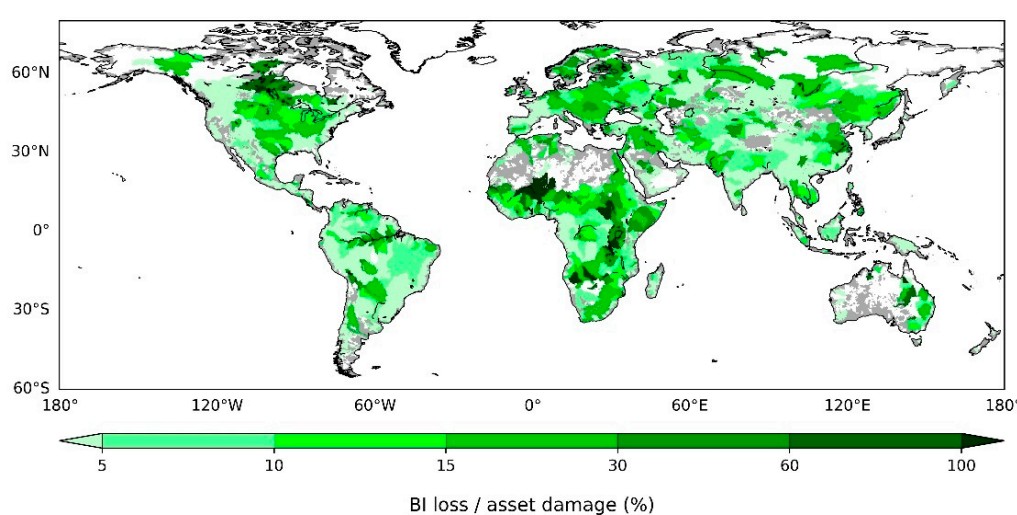

**Figure 6.** The percentage of BI loss to asset damage.

### 4.3. The Impact of Flood Protection on BI Loss Estimation

As shown in Figure 2, flood protection had a great effect on estimated BI loss. For example, in Malawi in 2012, the BI loss estimates with and without flood protection were USD 0.35 and 138 million (2005 PPP), respectively. To explore the effects of flood protection on estimated BI loss at the regional scale, we compared our reanalysis simulation results obtained with and without flood protection.

The global annual total BI loss and asset damage averaged from 1960 to 2013 without flood protection were USD 1864 and 3771 billion (2005 PPP), respectively. Flood protection led to BI loss and asset damage reductions of 98.6% and 96.5%, respectively. The ratios of BI loss to asset damage with and without flood protection, stratified according to continent, are shown in Figure 7. BI loss and asset damage were derived from annual average reanalysis simulation values. On all continents, the ratio of BI loss to asset damage decreased by an

average of 30% with and without flood protection; Australia and North America were consistent with the global average, whereas South America showed a 20% decrease. These results suggest that flood protection has a greater effect in reducing BI loss than in reducing asset damage.

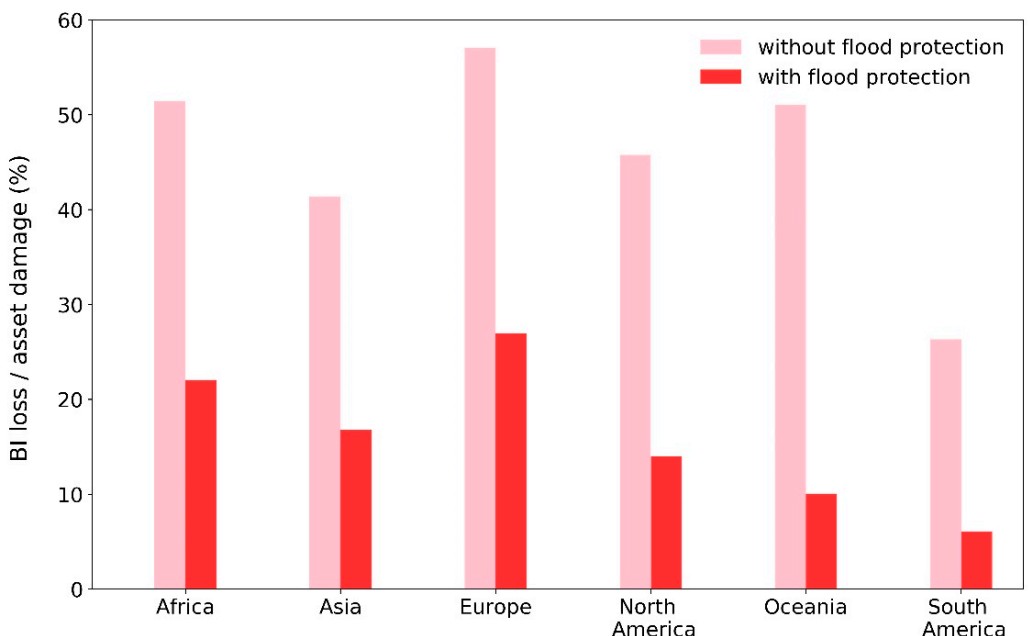

**Figure 7.** Ratio of BI loss to asset damage for each continent, with and without flood protection. Data are derived from annual averaged reanalysis simulation values.

To test this hypothesis, we targeted four river basins whose ratios of BI loss to asset damage were similar without flood protection (ca. 130%) but different with flood protection. Figure 8 shows the relationship between the ratio of BI loss to asset damage and the number of floods in targeted river basins (Yangtze, St. Lawrence, Senegal, and Niger). We defined the number of floods as the number of years in which the calculated annual maximum inundation depth exceeded the flood protection of the region at least once. The number of floods and the ratio of BI loss to asset damage were slightly lower in river basins with low flood protection level (FPL). For example, in Niger (FPL, 2 years), the number of floods slightly decreased from 54 to 40 with flood protection, while the ratio of BI loss to asset damage remained nearly constant, shifting from 143.2% to 121.8%.

In Senegal (FPL, 6 years), the reduction in the number of floods with flood protection was the same as the reduction in Niger, but the ratio of BI loss to asset damage decreased from 120.6% to 36.1%. In Senegal, FPL was sufficiently high to delay the day at which inundation depth exceeded FPL, although it did not prevent inundation, resulting in a decrease only in the ratio of BI loss to asset damage. Both the number of floods and the ratio of BI loss to asset damage decreased in basins with high FPL. In the Yangtze basin (FPL, 20 years), the number of floods significantly decreased from 54 to 4, while the ratio of BI loss to asset damage decreased from 131.3% to 33.0%. These results demonstrate that flood protection has a greater effect on BI loss than on asset damage; moreover, flood protection is important in the estimation of BI loss.

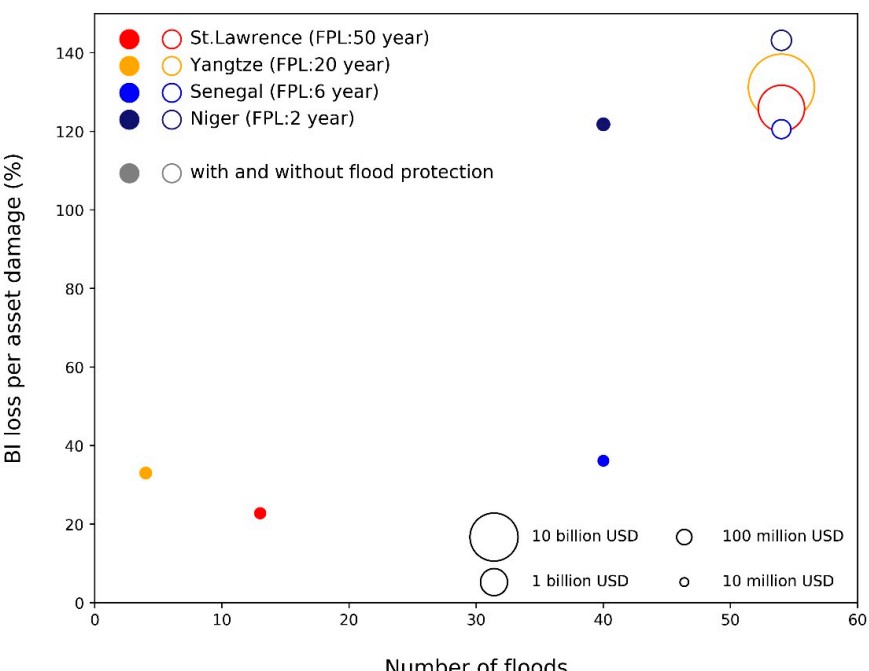

**Figure 8.** Relationship between the ratio of BI loss to asset damage and the number of floods exceeding flood protection in targeted river basins. Filled and empty circles indicate results with and without flood protection. Circle size is proportional to the sum of annual average asset damage and BI loss for each river basin.

### 4.4. Comparison of Future Estimated BI Loss and Asset Damage

The ratio of BI loss to asset damage tends to be greater in river basins with low FPL. However, our simulations showed that future ratios of BI loss to asset damage will be high even in large river basins with high FPL (Figure 9). Future BI loss to asset damage ratios exceeded 15% in most parts of Africa, where FPL is generally low. The BI loss to asset damage ratio is also expected to increase in most parts of China, India, and central Africa, as well as regions of Europe and America with high FPL. For example, the Mississippi River basin has an FPL of >50 years, but the BI loss to asset damage ratio is projected to increase by approximately 17.3% relative to past values. In contrast, the BI loss to asset damage ratio is projected to decrease in most parts of South America and Australia.

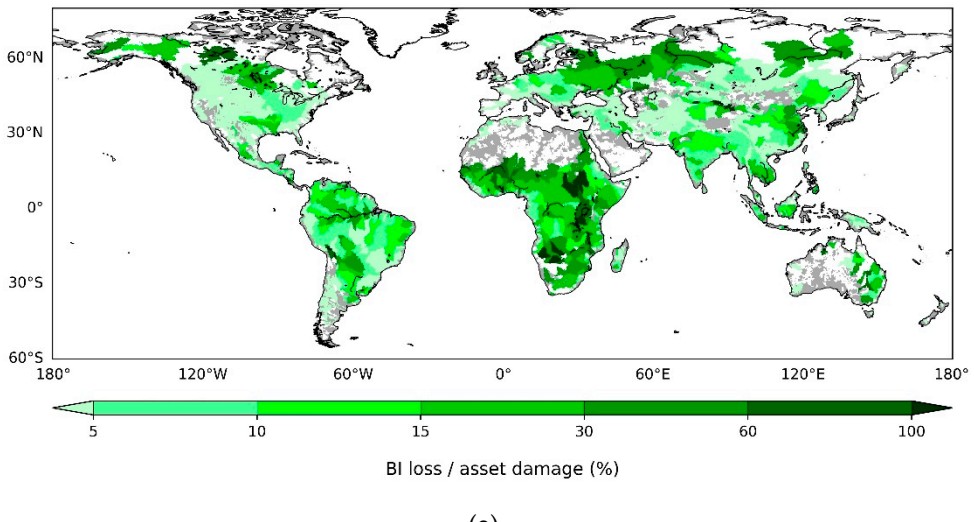

(**a**)

**Figure 9.** *Cont.*

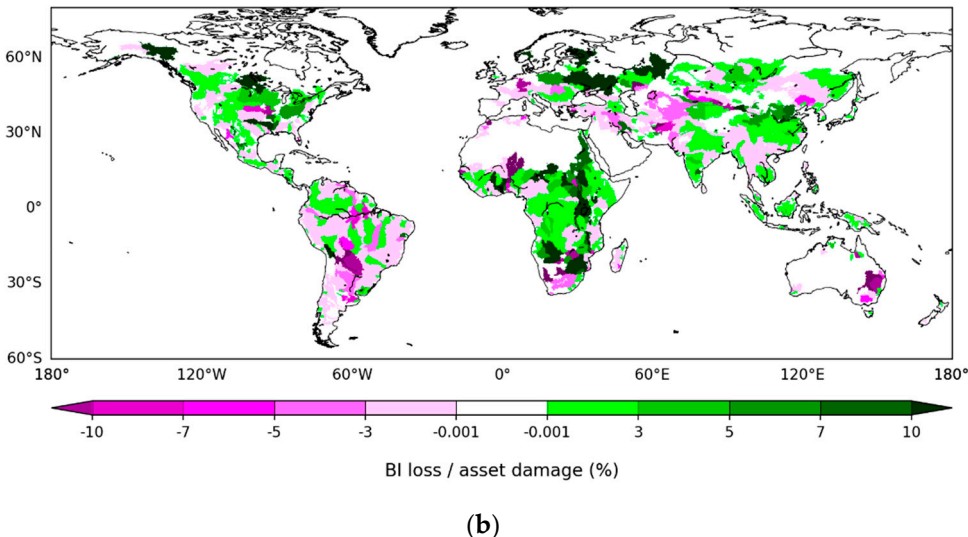

(**b**)

**Figure 9.** (**a**) BI loss to asset damage ratio averaged from 2061 to 2090 and (**b**) the difference in this ratio between past (1971–2000) and future study intervals for river basins with area $\geq$ 10,000 km$^2$. Gray areas indicate basins with area < 10,000 km$^2$.

The BI loss to asset damage ratio is projected to increase in river basins with high FPL presumably because of the increase in the number of floods with large return periods. As the flood return period increases, asset damage reaches a peak because the damage ratio reaches 1.0 at a specific inundation depth; in contrast, BI loss continues to increase during long inundation periods, resulting in an increase in the BI loss to asset damage ratio.

## 5. Conclusions

In this study, we developed a method to estimate flooding-related BI loss on a global scale. Estimated historical flood losses including BI loss showed magnitudes similar to losses reported by the World Bank. The estimated global BI loss and asset damage for the past study interval (1960–2013) were USD 26.9 and 130.9 billion (2005 PPP) per year, respectively.

BI loss tended to be greater in river basins with mild slopes such as the Amazon, which has long inundation periods. Future flood risk projection using the same framework under RCP8.5 and SSP3 scenarios showed increases in BI loss and asset damage per GDP by 0.32% and 1.78% (2061–2090 average), compared with values from the past interval (1971–2000 average), respectively. BI loss is projected to exceed 1.5% of the local GDP in some river basins with very flat terrains (such as the Nile, some river basins in Indonesia, and the Chao Phraya River basins), where floods will increase because of climate change.

In our historical simulation, the ratio of BI loss to asset damage was greater in river basins with FPL, which exhibited longer inundation periods and greater frequency of flood events than did regions with high FPL. This ratio is projected to increase even in river basins with high FPL, if flood magnitudes increase to the extent of basins such as the Mississippi River basin. Our results demonstrated that BI loss is not negligible in the estimation of global flood risk; notably, the projected risk changes with local FPL and flood magnitude.

The estimation values in this study might remain the scope of first order estimation due to the uncertainties of model, dataset, and methods. To conduct more realistic estimation of economic losses due to flood hazard, further improvements are needed especially on the estimation method of business interruption period and spatial distribution of assets. As mentioned before, we estimated these two parameters with data from limited cases in limited areas, and it is necessary to consider methods that can be applied more appropriately on a global scale. Although these limitations exist, we believe this study is useful since it provided the first estimate of BI loss due to flood on a global scale including its future change and discussed the relative importance of BI loss compared to the direct asset damage.

**Author Contributions:** Conceptualization, Y.H. and D.Y.; methodology, R.T. and M.T.; software, D.Y. and M.T.; validation, R.T.; writing—original draft preparation, R.T.; writing—review and editing, D.Y., Y.H. and M.T.; supervision, D.Y. and Y.H. All authors have read and agreed to the published version of the manuscript.

**Funding:** This research was supported by the Environment Research and Technology Development Fund (JPMEERF20202005) of the Environmental Restoration and Conservation Agency of Japan, a JSPS Grant-in-Aid for Scientific Research (18H01540), and the Integrated Research Program for Advancing Climate Models (TOUGOU) (JPMXD0717935457) from the Ministry of Education, Culture, Sports, Science and Technology (MEXT), Japan.

**Data Availability Statement:** Global river model CaMa-Flood model is available as open source software (http://hydro.iis.u-tokyo.ac.jp/~yamadai/cama-flood/ (accessed on 31 December 2021)). Data sources of statistical data used for loss estimation are summarized in Table 1.

**Conflicts of Interest:** The authors declare no conflict of interest.

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
