# Peer review of "Global-Scale Assessment of Economic Losses Caused by Flood-Related Business Interruption"

_water, doi:10.3390/w14060967_

Round 1
Reviewer 1 Report
The manuscript presents results in a organized and clear form, easy to read and understand. The methodology adopted is not new, but summarize results for diverse geographical areas and provide interesting commentss.
Minor typos need to be corrected: please, do a spell check.
Please, check the meaning of GFDRR, I believe it is Facility and not Faculty
I suggest publication, after minor corrections.
Reviewer 2 Report
Title
The Title reflects the paper’s content accurately.
Abstract
The Abstract determines the paper’s content and objectives in a very manifest and complete fashion.
- Introduction
Add the following in L30:
“As seen in [1] future projections for a 2 â—¦C global climate lead to the conclusion that blue water will show increasingly uneven runoff distribution, which, unless water storage infrastructure is increased, will turn into flood and flood occurrence is influenced by teleconnections as well shown by the 2010 Pakistan flood and the Russian heat wave, as well as in Iran’s Kan River basin, in the Southern Great Plains, the Missouri River Basin and the Yangtze River.“
In L36 flood risk is defined as “The risk from flooding to human and ecological systems is caused by the flood hazard (the frequency and/or magnitude of flood events), the exposure of the system affected (e.g. topography, or infrastructure in the area potentially affected by flooding) and the vulnerability of the system (e.g. design and maintenance of infrastructure, existence of early warning systems)” (attributed to your source quoted as [7] p.11-12)
In L37, after ‘countermeasures’ add “as it is seen that flood control is a major component of adaptation needs [2]”.
In L46 add “Risk is the expected loss (of lives, persons injured, property damaged, and economic activity disrupted) due to a particular hazard for a given area and reference period “ [3].
Otherwise, the Introduction is both adequate and highly informative.
- Materials and Methods
In L18-L183 you state the assumption that “Therefore, in this study, we assumed that building height within each grid was proportional to GDP per capita” based on a calculation that is quite simple and may not constitute a valid argument as it excludes parameters usually taken into consideration. The only connection found is in [4] Table 1 and in [5] p.15 “The stylized facts that we will establish later on show that, holding population constant, land area (hence k*), building heights ((H=L)(k)) and interior space consumption per household (h(k)) all increase directly with GDP per capita so that all three are larger in developed than developing cities.” which formally supports your statement.
In L244 add that BI is an indirect or consequential effect [6].
- Results
Very well worked out and presented including the validation process.
- Discussion
Quite exhaustive
- Conclusions
Precise and firmly based on the previous sections.
References
[1] Zisopoulou, K. and D. Panagoulia, “An In-Depth Analysis of Physical Blue and Green Water Scarcity in Agriculture in Terms of Causes and Events and Perceived Amenability to Economic Interpretation,” Water, vol. 13, no. 12, p. 1693, 2021, doi: 10.3390/w13121693.
[2] Zisopoulou, K., D. Zisopoulos, and D. Panagoulia, “Water Economics : An In-Depth Analysis of the Connection of Blue Water with Some Primary Level Aspects of Economic Theory I,” Water (Switzerland), vol. 14, 2022, doi: https://doi.org/10.3390/w14010103.
[3] U.N. Department of Humanitarian Affairs, Internationally agreed glossary of basic terms related to Disaster Management. Geneva, Switzerland: United Nations Department of Humanitarian Affairs, 2000. [Online]. Available: https://reliefweb.int/sites/reliefweb.int/files/resources/004DFD3E15B69A67C1256C4C006225C2-dha-glossary-1992.pdf
[4] Molinero, C. and S. Thurner, “How the geometry of cities explains urban scaling laws and determines their exponents,” J. R. Soc. Interface, vol. 18, pp. 1–18, 2019.
[5] Jedwab, R., P. Loungani, and A. Yezer, “How Should We Measure City Size ? Theory and Evidence Within and Across Rich and Poor Countries,” International Monetary Fund, Washington, D.C., U.S.A., WP/19/203, 2019.
[6] Messner, F. and V. Meyer, “Flood damage, vulnerability and risk perception - challenges for flood damage research,” in Flood Risk Management: Hazards, Vulnerability and Mitigation Measures NAIV, volume 67, J. Schanze, E. Zeman, and J. Marsalek, Eds. Leipzig, Germany: Springer, 2006, pp. 149–167. [Online]. Available: https://www.econstor.eu/bitstream/10419/45258/1/489068715.pdf
Reviewer 3 Report
Overall recommendation – moderate revision
The first question that immediately comes to a reader´s mind is as follows: Is a rigorous global-scale assessment of economic losses caused by flood-related business interruption possible at all? I am afraid that it is not. However, in my opinion the paper is worthy of interest.
In my view, the paper is kind-of half-baked meal and the authors have to humble and admit the obvious weaknesses (that can be justified in such a pioneering approach). Some assumptions are simplistic, yet quite innovative. It is easy to challenge them but there is no other way to get a first-order global view. Modelling results cannot be any better than the data and the assumptions. I would like the authors to identify caveats and to make more explanations.
Why were very old data used for (1961-2005, see lines 380-381) if the paper is submitted in 2022?
Detailed comments
Abstract and conclusions need attention. It is important to try to attract the audience who start contact with a paper from reading the abstract and the conclusions and only then – if these bits hold promise – may study the rest.
Abstract
Lines 11-12: “Flood risk estimation is an important component of land use management in regions prone to natural disasters.” This sentence, opening the abstract, is quite weak. In fact, flood-risk estimation is much more than a component of land use management
Lines 18-19: “global BI loss and asset damage of 26.9 and 52.4 billion USD per year, respectively, for 1960–2013.” Actually, a reader would like to know what USD are meant. Is it USD-2013 or inflation adjusted? Why does the study end in 2013?. It is worthwhile to explain this upfront.
I found „BI loss tended to be greater in river basins with mild slopes such as the Amazon, which has a long inundation period.” in lines 20-21 and 531. Also I found a similar statement on the Nile, Amazon and Chao Phraya three times – in lines 363, 416, 536. The authors should avoid copy-and-paste practices.
Lines 22-24: „Shared Socioeconomic Pathway 3 (SSP3) scenarios showed increases in BI loss and asset damage by 0.33% and 1.56% (2061–2090 average) compared with a past period (1971–2000 average), respectively”. This sentence in the Abstract discourages the reader who may not read the rest, because of mistrust. If the change was found of 0.33%, it is really meaningless, being far below the statistical error bound.
Lines 30-33: reference to UNIDRR needed
Lines 34-35: reference to Munich Re needed.
Actually the whole sentence „Munich-Re, a reinsurance company in Germany, estimated that global flood damage reached 27 billion USD in 2017” is suspicious! How can? This does not include the “wet” loss of the hurricanes Harvey and Maria.
Hurricane Harvey: Extreme rainfall producing historic flooding across Houston and surrounding areas. More than 30 inches of rainfall fell on 6.9 million people, while 1.25 million experienced over 45 inches and 11,000 had over 50 inches, based on 7-day rainfall totals ending August 31. This historic U.S. rainfall caused massive flooding that displaced over 30,000 people and damaged or destroyed over 200,000 homes and businesses. Total Estimated Costs: $125.0 ($138.8 – inflation adjusted to 2021 dollars) Billion
Hurricane Maria: Extreme rainfall up to 37 inches caused widespread flooding and mudslides across the island of Puerto Rico. Total Estimated Costs: $90.0 ($99.9) Billion
The authors may wish to consult a Europe-restricted paper by Kron et al. (2009), based on Munich Re perspective. There is a rule of thumb, that about half of wet-storm losses are associated with the impact of flooding, and the other half with other phenomena, such as wind.
Kron, W. et al. 2019 Reduction of flood risk in Europe - Reflections from a reinsurance perspective. Journal of Hydrology 576, 197-209.
Line 48: Global Faculty for Disaster Risk Reduction needs reference
Lines 50-51: “BI loss can be as extensive as asset damage in some instances, such as the 2011 Thailand floods, which caused 12 and 13.3 billion USD (2005 purchase power parity, PPP)”. Here, it would be interesting to discuss high-order propagation effects mentioned in the Abstract. This is, globally, quite an interesting case.
Lines 54-63: The authors may wish to consult a fine national study
Jiang Tong et al. (2020) Each 0.5°C of Warming Increases Annual Flood Losses in China by More than US$60 Billion. Bulletin of the American Meteorological Society (BAMS) Aug. 2020 E1464-E1474 141256
Lines 134-135: “Results from CaMa-Flood and MATSIRO have been used for global flood risk analysis in many studies [1,22].”. Two studies are not many!
Line 168, Line 229 – FLOPROS requires a reference here (# 14).
Lines 182-183 “we assumed that building height within each grid was proportional to GDP per capita”. This is a interesting, and probably disputable, assumption that deserves elaborating.
Line 192: “We used data from 11 countries, constituting 319 subnational data in total”. This is too little to meaningfully cover the whole Globe. Excessive extrapolation is made.
Line 268 “we assumed that the BI period was twofold longer than the inundation period.” – another assumption that could be challenged. The scientific basis of it is weak – just a few cases from Japan and USA. Yet, such a simplistic assumption can be made in a pioneering paper, because we cannot do any better. However, the authors should be more modest and humble, admitting that they made strong simplifying assumptions that could be challenged. Indeed extrapolation from two countries to a global scale cannot be rigorous.
Line 271: “asset damage was defined as the damage caused by the most severe flood in each year”. Why? What if there are several severe floods in one year, each causing asset damage.
Lines 273-275: Replace “Assets were assumed to be a constant multiple of GDP, a common practice in previous studies; they were calculated through the multiplication of GDP by the asset coefficient.” by “Assets were calculated through the multiplication of GDP by the asset coefficient, a common practice in previous studies.”
Lines 617 and 619 need more detailed references.
Conclusions are very short. The co-authors should extend this section considerably.
General – the readership deserves to get error bounds via uncertainty assessments. Actually, I wonder why an overambitious and nonrealistic objective of covering the whole world was made.
